# Examination and Modification of Multi-Factor Model in Explaining Stock Excess Return with Hybrid Approach in Empirical Study of Chinese Stock Market

## Jian Huang * and Huazhang Liu

Division of Business Management, Beijing Normal University-HongKong Baptist University United International College, Zhuhai 519087, China; spot_light@outlook.com or k530002087@mail.uic.edu.hk
* Correspondence: k530002046@mail.uic.edu.hk or jianhuang.951111@gmail.com; Tel.: +86-186-6454-9728

**Abstract:** To search significant variables which can illustrate the abnormal return of stock price, this research is generally based on the Fama-French five-factor model to develop a multi-factor model. We evaluated the existing factors in the empirical study of Chinese stock market and examined for new factors to extend the model by OLS and ridge regression model. With data from 2007 to 2018, the regression analysis was conducted on 1097 stocks separately in the market with computer simulation based on Python. Moreover, we conducted research on factor cyclical pattern via chi-square test and developed a corresponding trading strategy with trend analysis. For the results, we found that except market risk premium, each industry corresponds differently to the rest of six risk factors. The factor cyclical pattern can be used to predict the direction of seven risk factors and a simple moving average approach based on the relationships between risk factors and each industry was conducted in back-test which suggested that SMB (size premium), CMA (investment growth premium), CRMHL (momentum premium), and AMLH (asset turnover premium) can gain positive return.

**Keywords:** multi-factor model; risk factors; OLS and ridge regression model; python; chi-square test

## 1. Introduction

Financial markets are rife with uncertainties which make it difficult to forecast future market trends. Especially in the stock market, while providing investors with remarkable return, at the same time, it entails tremendous risk. As people pursue higher returns, they must bear the corresponding risks.

A pricing model with multiple factors is a promising approach to predict future stock prices. If the relationship between the risk and return can be expressed by multiple factors in a mathematical model, the standards of stock selection and corresponding investment strategies can be established. To be specific, each investor has their own risk preference, for instance, risk averse investors tend to bear lower risk and receive lower return. This specific type of investors has their preference in stock selection. In order to find suitable selection criteria, we need to quantify the relationship of risk premium factor and excess return. Investors can refer to the selection criteria to establish a corresponding trading strategy which can achieve their target excess return.

In the process of model examination, regression was conducted on single stocks which differed from the previous research which used portfolios. Especially, ridge regression was conducted instead of OLS regression. As for the process of model modification, two new factors were added to achieve higher explanatory power. Furthermore, we discussed the endogeneity and exogeneity for the risk premium factor. On the basis of economic objectives, we established a trading strategy for Chinese stock market.

Although previous research has worked well in the American stock market, these findings may be less practically applied to the Chinese market due the investor component. Since individual investors

contribute nearly 80% of the trading volume, investment behavior and preference can largely impact the market average return. However, the asymmetric information and investment concepts may lead to irrational behaviors. Therefore, we use risk premium factors to explain the excess return and the coefficients to measure the sensitivity of investor reactions to the risk premium.

To begin, we conducted single stock regression to examine the effectiveness of a five-factor model in Chinese stock market. Then we compare the coefficients of different factors under specific company types to discover the leading factor. We use the t-statistic to evaluate the significance of each factor. Regarding previous articles and research, we will add new factors in the model for better performance. Moreover, we conducted inter-factor cyclical research to study the pattern of factors. We can predict the rise and fall of the coefficient on a quarterly basis.

## 2. Literature Review

The multi-factor risk model has undergone a series of developments which can be divided into four major steps. At the beginning, in order to figure out the leading factors for security return, several researchers have contributed to the development of an asset pricing model. Initially, Markowitz (1952) proposed a mean-variance model to illustrate the statistical relationship between security risk and return in terms of standard deviation and expected rerun. It has established the foundation of modern finance theory. However, he had not specified the factors that explain expected security returns.

Based on the modern portfolio theory, the capital asset pricing model (CAPM) was developed by Sharpe (1964) and Lintner (1965) to illustrate the linear relationship between expected return and market risk premium.

$$\text{Ri} = \text{Rf} + \beta_{market}(\text{Rm} - \text{Rf}) \tag{1}$$

With the empirical tests in the stock market, they managed to find out the pattern of stock returns in line with the general stock market index. To be specific, the model estimates the relationships between the return on market index (the explanatory variable) and the return on the stock (the dependent variable). The regression coefficient of the single index model is referred to as beta which is a measure of the sensitivity of a stock to general movement in the market index. This empirical research symbolized the transition from qualitative analysis to quantitative analysis which also laid the foundation for the subsequent asset pricing models.

Since then, the CAPM model has been widely applied in research and empirical testing. However, with increasing abnormal returns which cannot be explained by existing factors, the market beta was no longer sufficient to describe expected return (Fama 1996). Therefore, Fama and French proposed a multifactor model consisting of three factors for market risk (Rm − Rf), market value, and book-to-market ratio (Fama and French 1993). In this model, Ri, Rf, and Rm stands for security expected return, risk free rate, and market return. SMB and HML are the risk premium factors. They illustrated that small stocks can generate higher returns than large stocks while value stocks can generate higher returns than growth stocks.

$$R_i - R_f = \text{a} + \beta_{market}\left(R_m - R_f\right) + \beta_{size}\text{SMB} + \beta_{BM}\text{HML} \tag{2}$$

It is also a supplement of arbitrage pricing theory (Ross 1976) which emphasizes that the expected return is not only affected by market risk but also a series of other factors. The generalized model (Bodie et al. 2014) illustrated the linear relationship of expected return and different factors (Bodie et al. 2017). In Equation (3), $F_j$ represents the factors and $b$ represents the coefficients.

$$r_i = a_i + \sum_{j=1}^{k} b_{ij}F_j + \epsilon_i, \; i = 1, 2\ldots, N \tag{3}$$

Moreover, Banz (1981) illustrated that return of securities are affected by several index including B/M ratio and E/P ratio which represent a series of risk premium. These articles largely contribute to the development of the multi-factor model.

In latter decades, the three-factor model was faced with a series of challenges. As the three-factor model was applied to stock trading and empirical testing, it appeared that some of the phenomenon cannot be explained by the model which can lead to unpredictable abnormal return. Therefore, more specific factors need to be added into the model to improve the accuracy. Novy-marx (2013) proved that profitability, measured by gross profits-to-assets, has roughly the same power as book-to-market value in explaining the average return. The following equation is based on the dividend discount model, $Y_{t+\tau}$ represents the earning for period $t + \tau$, $dB_{t+\tau}$ is the change in book equity, $r$ represents the expected return. It implied that higher market value leads to lower expected return, while higher earnings imply higher expected return.

$$M_t = \sum_{\tau=1}^{\infty} E(Y_{t+\tau} - dB_{t+\tau})/(1+r)^{\tau} \tag{4}$$

Aharoni et al. (2013) recorded an insignificant but statistically reliable relationship between investment pattern and average return. Other evidence also illustrated that the profitability and investment factors can explain some of the variation in average return.

Carhart (1997) conducted a study on the common factors in stock return and investment expense by adding momentum factor in three-factor model. It measures the tendency of price changes with a portfolio of long previous-12-month return winners and short previous-12-month loser stocks, which had an 8% accumulated yield. The momentum factor can explain 6.4% of the excess return.

Therefore, Fama and French (2015) added the two new factors, investment (CMA) and profitability (RMW), to build the five-factor model for measuring the effects of company size, valuation, profitability, and investment pattern in average stock returns. In this equation, they divide both sides by book value at time $t$ to create book-to-market ratio and present the relationship between return $r$ and valuation factor.

$$R_i - R_f = \text{a} + \beta_{market}(R_m - R_f) + \beta_{size}\text{SMB} + \beta_{BM}\text{HML} + \beta_{profitability}\text{RMW} + \beta_{investment}\text{CMA} + \varepsilon \tag{5}$$

Moreover, they use the RMW factor to represent the profitability and the CMA factor to represent investment. According to empirical study, the five-factor model has a higher effectiveness than the three-factor model, which can explain 71–94% of the variation of average return. In the test, researchers divided all the stocks with three sets of factors. The result also implied that book-to-market factor is redundant for describing average return using the American stock data from 1963 to 2013. Also, the model fails to capture the low average return for small firms with low profitability and high investment.

Sehgal and Vasishth (2015) tested the model in various emerging markets and discovered that the change of price and trading volume are partly risk based and partly behavioral. The research indicated that behavioral factors are necessary in the study of the multi-factor model. Except for the existing risk premium factors in Fama and French model, Peng et al. (2014) studied the effect of different investment sentiments on the market return from which they found out customer satisfaction is a significant factor for abnormal return. Moreover, a human capital factor was considered in terms of compensation level (Moinak and Balakrishnan 2018). Zahedi and Rounaghi (2015) applied an artificial neural network to assess the components of a multi-factor model and predict future stock prices.

Considering the features of data and the time-varying factors, a range of studies applied various methods to address problems via multi-factor models. Akter and Nobi (2018) examined the distribution and frequency distribution for both daily stock returns and volatility. Furthermore, Chen and Kawaguchi (2018) distinguish two significant regimes (a persistent bear market and a bull market) to examine market time-varying risk factors to achieve Markov regime-switching. These studies examined the model specifically in certain periods which enables the researchers to compare the model performances in different situations which can improve the applicability of the model.

Some researchers conducted modification on the basic model with specific structure. Ronzani et al. (2017) suggested that β (systematic risk) evolves over time and the model with time-varying β provide less

conservative VaR measures than the static β. While Cisse et al. (2019) examined the dynamics of the model with Kalman filter and Markov switching (MS) model and proved that the former method fits better in the model. Bhattacharjee and Roy (2019) proposed a social network dependence structure to address such misspecifications. For the investment aspect, Frazzini et al. (2013) suggesting that Buffett's returns are more due to stock selection than to his effect on management.

## 3. Methodology

### 3.1. Hypotheses

**Hypothesis 1.** *Market premium has a positive relationship with excess return.*

**Hypothesis 2.** *Size premium has a positive relationship with excess return.*

**Hypothesis 3.** *Book-to-market premium has a positive relationship with excess return.*

**Hypothesis 4.** *Profitability premium has a positive relationship with excess return.*

**Hypothesis 5.** *Investment growth premium has a positive relationship with excess return.*

**Hypothesis 6.** *Momentum premium has positive a relationship with excess return.*

**Hypothesis 7.** *Asset turnover premium has a positive relationship with excess return.*

Hypotheses are conducted on the relationship between dependent variable and independent variables. To be specific, the positive relationship stated that risk premium factor can generate higher excess return. For instance, a company with higher profitability can achieve higher returns than a low profitability company. Since previous researchers have not examined the effect of factors on single stocks, we decided to retest the effect of seven factors on excess return for each security and summary by industries.

### 3.2. Research Design

The data were collected from the Choice, Tongdaxin, and Resset database. Choice and Tongdaxin can provide historical values of stock financial ratios and Resset provides the basic information of companies and interest rates. The samples are 1097 stocks in the Chinese stock market, including Shanghai securities exchange market and Shenzhen securities exchange market from, 2007 to 2018 because large quantity of firms transformed their non-tradable shares into tradable shares in the process of Chinese Reformation of National Owned Stock from 2005 to 2006 and the time range after 2007 provides the research with more companies' data. The type of data is quarterly data which includes 47 quarterly data points for each stock. In the process of data cleaning, those stocks with missing data will be deleted in the samples.

Even if Fama and French (2015) had delivered the five-factor model, yet they only consider the factor performance in different groups of portfolio but not the factors' effect on each security in the stock market. Thus, by applying variable-intercept models mentioned in the book of Hsiao (2003), we ran the panel data for each stock to test the effect of these five premiums in the Model 1. The study bridged the relationships between the excess return of each security and different premium factors via OLS (ordinary least squares regression), which means each stock has their individual coefficient and there are more than 1000 regression functions (Johnson and Wichern 2008). During the research process, this paper classified the securities into different groups to calculate risk premium factors based on market value, book-to-market ratio, ROE, and the growth of investment. Then, this research considered the average coefficients' level, average significant level, and their distribution.

In Model 2, the momentum factor and turnover factor are added into the model. We developed an innovative method to study each single stock in the market, in order to find out the general pattern of the stock market. In contrast to the former ways that examine the model with the diversified portfolio, we conducted ridge regression on single stocks and divided the stocks in industries. The 28-industry classification standards are from ShenWan industry index.

Figure 1 illustrates the research structure and their corresponding functions for the results. The research includes two major parts, which are multi-factor examination with single stock regression and time-series analysis for risk factors.

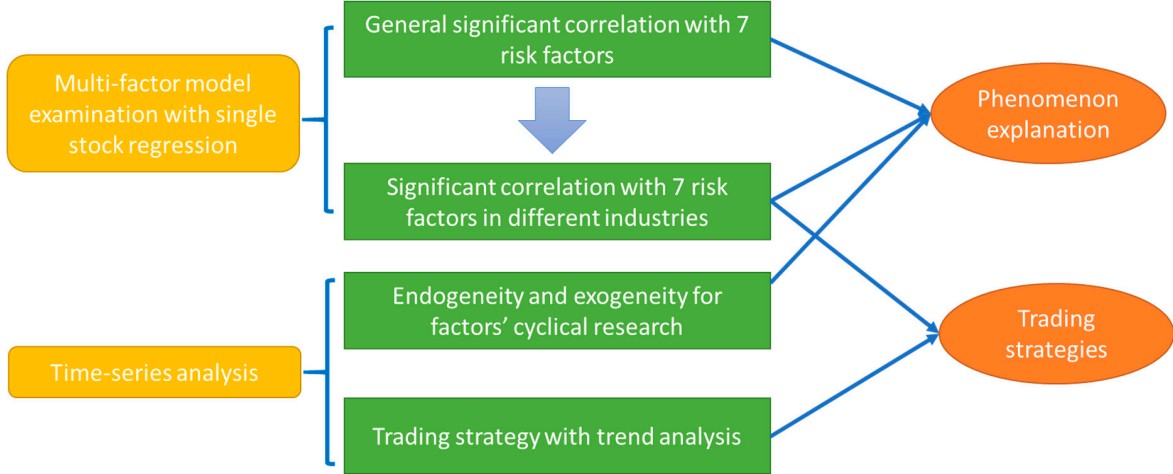

**Figure 1.** Multi-factor model examination with single stock regression and time-series analysis.

The first part involved stability test, OLS regression, ridge regression, and robustness test to find the significant correlations between stock excess return and seven risk factors. In addition, a chi-square test was conducted to examine whether the effect of factors various in different industries. Moreover, by measuring the percentage of positive and negative correlations in each industry, the significant relationships between factors and industry was discovered.

The second part of research covered chi-square test to find out the endogeneity and exogeneity (or called pattern of fluctuation) for seven risk factors and a back test of trading strategy with trend analysis. For details, for chi-square testing, we firstly recorded the rise and fall pattern in the neighboring quarters. There were four kinds of pattern (rise after rise, rise after fall, fall after rise, fall after fall). We calculated each patterns' amount and conducted chi-square testing on the pattern of each factor. If a factor passes the test, it can be inferred that the current pattern is in accordance with the previous period. Therefore, with the endogeneity and exogeneity of the factors, we can predict the future rise and fall based on the current pattern. Moreover, a trading strategy was established based on the trend analysis of factors. As for the investment portfolio, stocks were selected according to the effect of seven factors on industries. A simple back test was also conducted to examine the performance of the trading strategy.

The result of general significant correlations and the factors' effect in various industry can be used in some economic phenomenon explanation. Furthermore, combing the significant relationships between risk factors and corresponding industries with trend analysis, trading strategies can be built.

Factor Selection

We modified the multi-factor model to adapt for the Chinese stock market. Since some of the existing factors in the model have poor performance in the regression analysis and empirical test, it is significant to add or delete some of the factors to improve the model.

In accordance with previous studies, we selected two new factors (momentum factor and turnover factor). Firstly, the momentum effect is remarkable in the medium term of stock market. Momentum

effect was proposed by Jegadeesh and Titman (1993) to illustrate the phenomenon that stock performed well in the past is more likely to achieve higher return. Regarding the composition of investor and investment behavior in stock market, we suggested that the momentum effect can explain the part of the excess return. Secondly, we added a turnover factor into the model. Asset turnover rate is a component of DuPont analysis, which is used to analyze the return of shareholder's equity (Wild 2016). Asset turnover rate is a vital measure of a company's operation capacity. A higher asset turnover rate indicates that a company has better operation capacity and higher efficiency. When conducting value investing, this financial ratio is primarily regarded before investment, as it can directly reflect the condition of a company, for example profitability, operation capacity, and so on.

### 3.3. Research Process

3.3.1. Multi-Factor Model Examination with Single Stock Regression Analysis

- **Stability Test**

  DV: Ri-Rf
  IV: Rm-Rf, SMB, RMW, HML, CMA, CRMHL, AMLH
  CV: Time, Time^2, Season

  If the three control variables which refers to time, the square of time and season can pass the *t*-test with high p-level, these three variables should be put into the regression models, otherwise they can be deleted.

- **OLS Regression**

  We conducted OLS regression on 47 quarters for each single stock, in order to examine the multi-factor model. According to OLS, to calculate the beta (coefficient) of each independent variable, the matrix operation combines X and y. In this formula, X is a $47 \times 7$ matrix, in which 47 is the 47 time-series of data and 7 represents seven proposed risk factors. The y is a $47 \times 1$ matrix and 1 indicates the excess return of a single stock. Since there are 1097 stock, X matrix is fixed yet y matrix is the data of different stocks.

$$\beta = (X^TX)^{-1}X^Ty \tag{6}$$

$$
\begin{bmatrix} \beta_1 \\ \beta_2 \\ \vdots \\ \vdots \\ \beta_7 \end{bmatrix} = \left( \begin{bmatrix} SMB_1 & RMW_1 & \dots & AMLH_1 \\ \vdots & \vdots & & \vdots \\ \vdots & \vdots & & \vdots \\ SMB_{47} & RMW_{47} & \dots & AMLH_{47} \end{bmatrix}^T \begin{bmatrix} SMB_1 & RMW_1 & \dots & AMLH_1 \\ \vdots & \vdots & & \vdots \\ \vdots & \vdots & & \vdots \\ SMB_{47} & RMW_{47} & \dots & AMLH_{47} \end{bmatrix} \right)^{-1} \begin{bmatrix} SMB_1 & RMW_1 & \dots & AMLH_1 \\ \vdots & \vdots & & \vdots \\ \vdots & \vdots & & \vdots \\ SMB_{47} & RMW_{47} & \dots & AMLH_{47} \end{bmatrix}^T \begin{bmatrix} y_1 \\ y_2 \\ \vdots \\ \vdots \\ y_7 \end{bmatrix} \tag{7}
$$

  We used computer simulation in Python, which can help us automatically run regression 1097 times. The python program would print out the result of regression and calculate the number of mean value of coefficients and *t*-value, which were collected in a $7 \times 1097$ matrix. We also drew the frequency histogram for each factor to study the distribution of the coefficients. Based on these assumptions, the majority of the coefficients should be positive. If most of them come out negative, we may infer that the five-factor model is not suitable for the Chinese stock market.

- **Ridge Regression**

  Since we added two more factors in the model, we applied ridge regression instead of OLS regression. On one hand, ridge regression can prevent an over-fitting result from extra factors. On the other, it can prevent multi-collinearity and increase the significance of the factors. Ridge regression is developed based on OLS regression by adding regularization term $\lambda I$. The regularization term can discover the factor with collinearity and force the coefficient to approach zero to ensure that the effect of multi-collinearity is minimized. To calculate the beta (coefficient) of each independent variable,

the matrix operation combines X, y, $\lambda$, and I. X and y are the same as the matrix in the OLS model. $\lambda$ is an optimization hyperparameter and I is a $7 \times 7$ identity matrix.

$$\beta = (X^T X + \lambda I)^{-1} X^T y \tag{8}$$

$$\begin{bmatrix} \beta_1 \\ \beta_2 \\ \vdots \\ \beta_7 \end{bmatrix} = \left( \begin{bmatrix} SMB_1 & RMW_1 & \cdots & AMLH_1 \\ \vdots & \vdots & & \vdots \\ \vdots & \vdots & & \vdots \\ SMB_{47} & RMW_{47} & \cdots & AMLH_{47} \end{bmatrix}^T \begin{bmatrix} SMB_1 & RMW_1 & \cdots & AMLH_1 \\ \vdots & \vdots & & \vdots \\ \vdots & \vdots & & \vdots \\ SMB_{47} & RMW_{47} & \cdots & AMLH_{47} \end{bmatrix} + \begin{bmatrix} \lambda & 0 & 0 & 0 \\ 0 & \lambda & 0 & 0 \\ \vdots & \vdots & \ddots & \vdots \\ 0 & 0 & 0 & \lambda \end{bmatrix} \right)^{-1} \begin{bmatrix} SMB_1 & RMW_1 & \cdots & AMLH_1 \\ \vdots & \vdots & & \vdots \\ \vdots & \vdots & & \vdots \\ SMB_{47} & RMW_{47} & \cdots & AMLH_{47} \end{bmatrix}^T \begin{bmatrix} y_1 \\ y_2 \\ \vdots \\ y_7 \end{bmatrix} \tag{9}$$

When we applied the ridge regression, we need to find a $\lambda$ that can provide the largest sum of R-square which are the number of positive factors and percentage of positive coefficient. We set the summary of these three measures as our target function to find out the optimal $\lambda$ with iteration.

$$\lambda = \text{argmax}(Q) \tag{10}$$

Q = P average value + number of positive coefficients + R square mean + number of variables whose $p$ value over 0.9 (9).

- **Robustness Test**

In this part, a zero-mean test for OLS and ridge regression model is adopted to check the robustness of models. Both OLS and ridge regression models are fixed-coefficient models, which means that they have an assumption that the covariance of errors between different units (stocks) are equal to zero. Thereafter, a residual covariance matrix between 1097 stocks' regression can be drawn by Python and calculate the mean and standard deviation of all numbers in covariance matrix. Then a $t$-test is conducted to test whether $E(u_{i,j})$ is equal to zero.

Covariance matrix of residual

$$\begin{bmatrix} u_{1,1} & u_{1,2} & \cdots & u_{1,1097} \\ u_{2,1} & \ddots & \cdots & \vdots \\ \vdots & \vdots & \ddots & \vdots \\ u_{1097,1} & \cdots & \cdots & u_{1097,1097} \end{bmatrix} \tag{11}$$

$u_{1,2}$ means the covariance between stock 1 and 2.

$$\text{T-value} = \frac{E(u_{i,j})}{STDE(u_{i,j})} \tag{12}$$

### 3.3.2. Time-Series Analysis for Risk Factors

- **Chi-Square Test**

We use chi-square in two steps respectively. Firstly, it was used to examine the different effect of factors in different industries. In the test we divided the significance of factors based on industries and calculated the total value $\chi^2$.

Secondly, it was applied to find out the pattern in the inter-factor direction prediction analysis. We divided the pattern into four types with a combination of increase and decrease. Then we used chi-square to find out whether there is a pattern for factor change direction. With the increase and decrease pattern, we can predict the next quarter movement based on the current situation. As we can see in Figure 2, suppose "0" represents the increase of risk factor in one term (or one season) and "1" represents the decrease of risk factors.

$$R = \text{Max}(F1, F2) + \text{Max}(F3, F4) \tag{13}$$

$$W = \text{Min}(F1, F2) + \text{Min}(F3, F4) \tag{14}$$

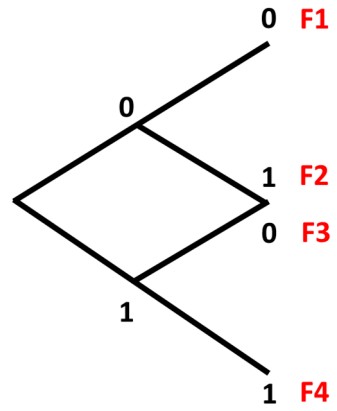

**Figure 2.** Pattern of factor's change.

In Figure 3, if investors spontaneously make investment decision, the frequencies of right and wrong decisions are equitable. Both are T/2. (T is number of total transaction time)

**H0**: *There is no relationship between right or wrong investment decisions and direction prediction rules.*

**H1**: *There is a relationship between right or wrong investment decisions and direction prediction rules.*

|  | Follow the rule | Randomly select | Total |
|---|---|---|---|
| Right investment decision | $R$ | $T/2$ | $R+T/2$ |
| Wrong investment decision | $W$ | $T/2$ | $W+T/2$ |
| Total | $R+W$ | $T$ | $R+W+T$ |

**Figure 3.** Observed frequencies: $f_o$.

In Figure 4, the expected frequency of right decisions ($R_e$) and wrong decisions ($W_e$) made by the rule can be calculated by Equations (15) and (16). Also, the expected frequency of right or wrong decision made by random selection ($(T/2)_e$) can be calculated by Equation (17).

$$R_e = (R + W) \times (R + T/2)/(R + W + T) \tag{15}$$

$$W_e = (R + W) \times (W + T/2)/(R + W + T) \tag{16}$$

$$(T/2)_e = T \times (R + T/2)/(R + W + T) \tag{17}$$

|  | Follow the rule | Randomly select |
|---|---|---|
| Right investment decision | $R_e$ | $(T/2)_e$ |
| Wrong investment decision | $W_e$ | $(T/2)_e$ |

**Figure 4.** Expected frequencies: $f_e$.

According to the significant table (Figure 5) of chi-square test, if total chi-square level is >2.706, we have 90% confidence to reject the null hypothesis (H0). If total chi-square level is >3.841, we have 95% confidence to reject the null hypothesis (H0). If total chi-square level is >6.635, we have 99% confidence to reject the null hypothesis (H0).

$$\text{Total chi-square level} = (R - R_e)^2/R_e + (W - W_e)^2/W_e + 2[T/2 - (T/2)_e{}^2]/(T/2)_e \tag{18}$$

$$\text{The degree of freedom} = (\text{column} - 1)(\text{row} - 1) = (2 - 1)(2 - 1) = 1 \tag{19}$$

| | Follow the rule | Randomly select |
|---|---|---|
| Right investment decision | $(R - R_e)^2/R_e$ | $[T/2 - (T/2)_e]^2/(T/2)_e$ |
| Wrong investment decision | $(W - W_e)^2/W_e$ | $[T/2 - (T/2)_e]^2/(T/2)_e$ |

**Figure 5.** Chi-square level: $\chi^2 = (f_o - f_e)^2 f$.

- **Back-Test for Trading Strategy**

After factors' cyclical research and trend analysis for fluctuation of risk factors, the trading strategy based on moving average approach and correlations between factors and industries can be formed. By deciding the time to do the transactions and stocks which should be bought or sold, the float return from 2007S4 to 2018S3 and annual expected return can then be calculated.

*3.4. Assumptions for Multi-Factor Examination*

The following assumptions are made in applying the multi-factor model:

1. Perfect market: there are no tax and transaction costs.
2. People are risk averse or rational.
3. People can lend or borrow money at a risk-free rate freely.
4. There is a trade-off between risk and return for all securities.
5. Everyone can obtain market information equally and freely.
6. Investors have the same expectations for this market.

The intuition and assumption behind the hypotheses are presented in the Appendix A with measurements of factors and graphs.

*3.5. Models and Variable Definitions*

Model 1:

$$R_i - R_f = \text{a} + \beta_{market}\left(R_m - R_f\right) + \beta_{size}\text{SMB} + \beta_{BM}\text{HML} + \beta_{profitability}\text{RMW} + \beta_{investment}\text{CMA} + \varepsilon$$

Model 2:

$$R_i - R_f = \text{a} + \beta_{market}\left(R_m - R_f\right) + \beta_{size}\text{SMB} + \beta_{BM}\text{HML} + \beta_{profitability}\text{RMW} + \\ \beta_{investment}\text{CMA} + \beta_{momentum}\text{CRMHL} + \beta_{turnover}\text{AMHL} + \varepsilon$$

The details of the factors (Rm-Rf, SMB, HML, RMW, CMA, CRMHL, AMHL) including explanation and graphs are presented in Appendix A.

Table 1 displays the dependent variable, independent variables and the corresponding labels and explanations.

**Table 1.** Dependent variable and independent variables.

| Name | Label | Note |
|---|---|---|
| | **DV** | |
| Excess return | $R_i - R_f$ | The return of security mines risk-free rate of return |

**Table 1.** *Cont.*

| Name | Label | Note |
|------|-------|------|
| **IV** | | |
| Abnormal return | a | The constant term of formula |
| Market premium | $R_m - R_f$ | The return of market index (in this model, market index is Shanghai stock exchange market index) mines risk-free rate of return |
| Size premium (Small minus Big) | SMB | The return on a diversified set of small stocks minus the return on a diversified set of big stocks. |
| Book-to-market premium (High minus low) | HML | The difference between the returns on diversified portfolios of high and low B/M stocks. |
| Profitability premium (Robust minus weak) | RMW | The difference between the returns on diversified portfolios of stocks with robust and weak profitability. |
| Investment growth premium (Conservative minus aggressive) | CMA | The difference between the returns on diversified portfolios of the stocks of low and high investment firms, which we label as conservative and aggressive. |
| Momentum premium (High momentum minus low momentum) | CRMHL | The difference between higher momentum (higher accumulated return) companies' average return and lower momentum (lower accumulated return) companies' average return in a diversified portfolio or in the market. |
| Asset turnover premium (Low turnover rate minus high turnover rate) | AMLH | The difference between low asset turnover companies' average return and higher turnover companies' average return or in the market. |

## 4. Results

*4.1. Multi-Factor Model Examination with Single Stock Regression Analysis*

4.1.1. Stability Test

DV: Ri-Rf
IV: Rm-Rf, SMB, RMW, HML, CMA, CRMHL, AMLH
CV: Time, Timeˆ2, Season

According to the result of stability test (presented in Appendix E), the control variable including Time, Timeˆ2, Season did not pass the *t*-test, the significance level is far less than 0.9. Thus, they are removed from the model. It means that excess return is not affected by time.

4.1.2. Regression Models

- **OLS Regression**

**Model 1:** Five-factor model

DV: Ri-Rf
IV: Rm-Rf, SMB, RMW, HML, CMA

**Model 2:** Seven-factor model

DV: Ri-Rf
IV: Rm-Rf, SMB, RMW, HML, CMA, CRMHL, AMLH

**Model 3:** Modified optimal model

　　DV: Ri-Rf
　　IV: Rm-Rf, RMW, HML, CMA, CRMHL, AMLH

- **Ridge Regression**

**Model 4:** Ridge regression model

　　DV: Ri-Rf
　　IV: Rm-Rf, SMB, RMW, HML, CMA, CRMHL, AMLH

We conducted OLS regression on Model 1, 2, and 3, and ridge regression on Model 4. After conducting the regression analysis, we recorded each factors' coefficients and mean of R square. We created frequency histograms to observe the distribution of coefficients. The graphs provide a general view of the coefficients.

In Table 2, above each histogram, there are two numbers, mean of coefficients and the proportion of coefficients that passed the *t*-test (in the parenthesis).

To be specific, we divided the coefficients into uncorrelated, positively correlated, and uncorrected, based on 95% confidence interval. Table 3 shows the percentage of coefficient. SMB, CMA, Rm-Rf, CRMHL, and AMHL have more positive coefficients. Whereas RMW and HML have more negative coefficients.

In ridge regression, we obtain the λ by calculating the maximum of the objective function Q. When λ equals to 0.008, the result of ridge regression is maximized. In Figure 6, we can find out the largest Q value at 0.008.

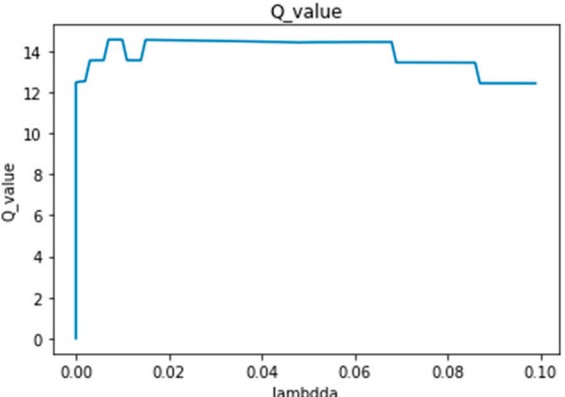

**Figure 6.** Target function optimization.

Some factors have negative effects on the excess return, which deviates from our assumption. Moreover, regarding the factors that have an even distribution. We need to classify them in industry groups to discover a further pattern.

**Table 2.** R square mean, coefficients' mean and p-level for four models.

| | | OLS | | | Ridge Regression |
|---|---|---|---|---|---|
| | | **Model 1** | **Model 2** | **Model 3** | **Model 4** |
| **Mean of R square** | | 0.5692 | 0.6078 | 0.5911 | 0.6048 |
| **Mean of coefficient (p-level)** | Rm-Rf | −0.29 (0.9535) | 0.29 (0.9991) | 0.82 (0.9991) | 0.21 (0.9991) |
| | SMB | −0.29 (0.8669) | 0.34 (0.1641) | | 0.29 (0.8368) |
| | RMW | −0.74 (0.9189) | 0.24 (0.9891) | 0.40 (0.9763) | 0.22 (0.9298) |

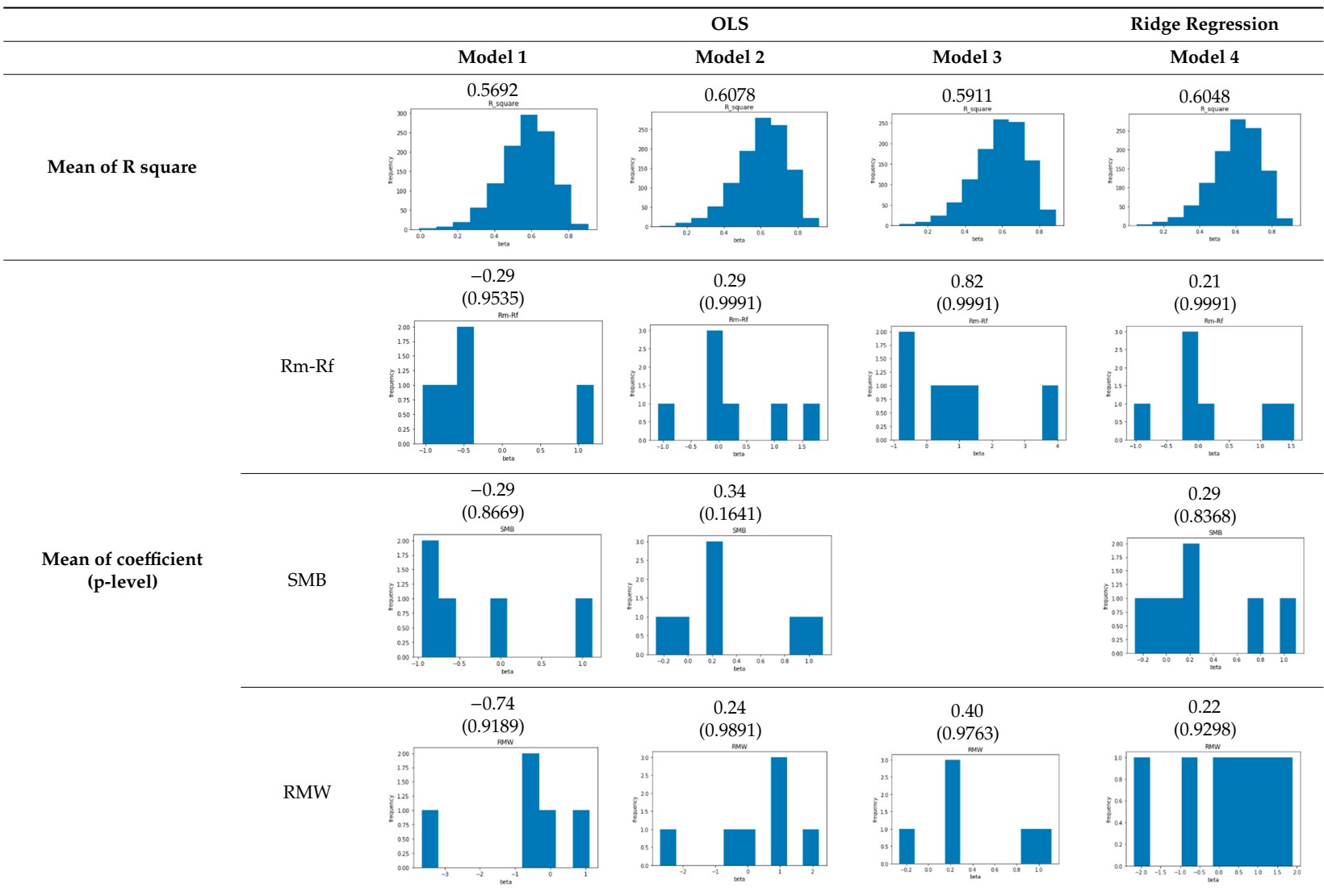

**Table 2.** *Cont.*

| | | OLS | | | Ridge Regression |
|---|---|---|---|---|---|
| | | **Model 1** | **Model 2** | **Model 3** | **Model 4** |
| **Mean of coefficient (p-level)** | HML | 0.13 (0.3874) | 0.30 (0.9690) | −0.27 (0.9444) | 0.22 (0.9681) |
| | CMA | −0.11 (0.9335) | 0.58 (0.9900) | −0.50 (0.9763) | 0.38 (0.9909) |
| | CRMHL | | −0.14 (0.9617) | 0.33 (0.9243) | 0.02 (0.9535) |
| | AMLH | | −0.04 (0.9991) | −0.72 (0.9991) | 0.06 (0.9991) |

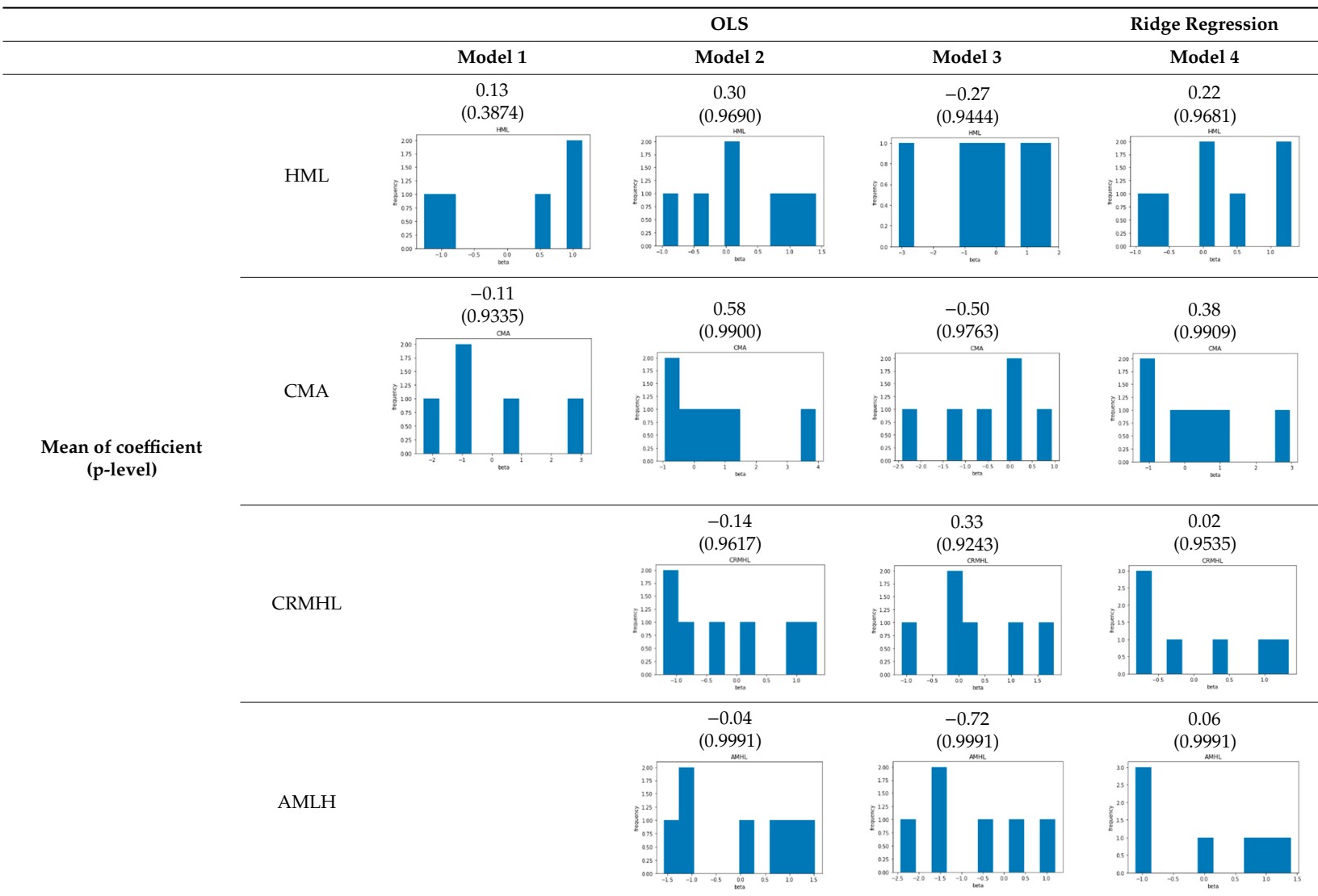

**Table 3.** Correlations between seven factors and excess return of stocks.

|  | SMB | RMW | HML | CMA | AMLH | CRMHL | Rm-Rf |
|---|---|---|---|---|---|---|---|
| Positive | 61.0% | 24.2% | 23.0% | 56.2% | 62.4% | 77.8% | 99.8% |
| Negative | 22.7% | 68.7% | 73.8% | 42.8% | 37.5% | 17.6% | 0.1% |
| Uncorrelated | 16.3% | 7.0% | 3.2% | 0.9% | 0.1% | 0.1% | 0.1% |

### 4.1.3. Robustness Test: Zero Mean Residual Testing

According to Table 4, since *t*-values in four models are less than 2.021, we have 95% confidence not to reject the null hypothesis, which means the residuals are randomly independent for all stocks, both the OLS and ridge regression model pass the zero mean residual test.

**Table 4.** Zero mean residual testing.

|  | OLS *t*-Value |  | Ridge Regression *t*-Value |
|---|---|---|---|
| Model 1 | Model 2 | Model 3 | Model 4 |
| 0.5811 | 0.5393 | 0.5416 | 0.5387 |

### 4.1.4. Industry Analysis

According to the result of chi-square test, each industry corresponds differently to the factors. Therefore, effect of factors needs to be discussed based on different industries respectively. The details of the test results are presented in the Appendix B.

We classify the stocks in 28 industries. Also, we divided the coefficients in positive correlation and negative correlation. As for data cleaning, we only preserved the factors that contained over 70% coefficients are of the same signs. The specific table is presented in the Appendix C.

In the Table 5, the industries are classified in manifold groups based on the significance level and the relationship with each factor.

According to the table, there is no industry that can be explained by seven factors simultaneously. The banking and steel industries are best fitted in the seven-factors model, since they have six significant factors and the highest sum of percentage of significant coefficients.

Each factor has different relationships with different industries. For market premium (Rm-Rf), it can apply to all industries. For SMB, it is positively related with seven industries (Group A), yet SMB is negatively correlated with Steel industry. With respect to RMW, it has negative relationships with most industries (Group D). However, RMW has significant positive relationships with bank industry. HML is negative related to multiple industries (Group F). Also, HML has significant positive relationships with the banking industry. CMA is positively correlated with five industries (Group G). Meanwhile, CMA is negatively correlated to three industries (Group H) including extractive, banking, and nonferrous metal. CRMHL is positive correlated with most of industries (Group I) except media, electrical equipment, non-bank finance, animal husbandry and fishery, commercial, and comprehensive industries. AMLH has a negative correlation with Group K. At the same time, AMLH is positively related with three other industries (Group J) including housing, non-bank finance, and banking.

Moreover, the industries with the same pattern are regarded as coordinated industry. Additionally, the industries with different pattern are regarded as deviated industries. To be specific, the banking industry is considered as a deviated industry since it has a positive RMW and HML factor, while other industries have negative factors. The graph shows four groups of coordinated industry—including telecommunication and electronics, chemistry and automobile, light manufacturing and defense, and nonferrous metal and extractive—while banking is regarded as a deviated industry.

Considering the trend of the factors, we can formulate corresponding strategies. To be specific, based on the condition of each factor and the classification from Group A to L which consist of different industries presented in Table 5, we formulated manifold strategies in Table 6.

**Table 5.** Relationships of industries and seven factors.

| | Positive Relationships | Negative Relationships |
|---|---|---|
| **SMB** | (Group A: 7)<br>Electrical Equipment, Electronics, Computer, Construction Material, Food, Telecommunication, and Leisure Service | (Group B: 1)<br>Steel |
| **RMW** | (Group C: 1)<br>Banking | (Group D: 16)<br>Extractive, Media, Electrical Equipment, Textiles & Garments, Steel, Defense, Chemistry, Mechanical Equipment, Computer, Construction Material, Transportation, Automobile, Light Manufacturing, Leisure Service, Nonferrous Metal, and Comprehensive Industries. |
| **HML** | (Group E: 1)<br>Banking | (Group F: 19)<br>Extractive, Media, Electrical Equipment, Electronics, Housing, Textiles & Garments, Steel, Defense, Chemistry, Mechanical equipment, Computer, Animal Husbandry and Fishery, Automobile, Light Manufacturing, Commercial, Telecommunication, Leisure Service, Medical, Nonferrous Metal, and Comprehensive Industries. |
| **CMA** | (Group G: 5)<br>Textiles & Garments, Utilities, Defense, Light Manufacturing, and Commercial | (Group H: 3)<br>Extractive, Banking, and Nonferrous Metal |
| **CRMHL** | (Group I: 20)<br>Extractive, Electronics, Housing, Textiles & Garments, Steel, Utilities, Defense, Chemistry, Mechanical Equipment, Computer, Domestic Appliance, Construction material, Transportation, Automobile, Light Manufacturing, Telecommunication, Leisure Service, Medical, Banking, and Nonferrous Metal. | |
| **AMLH** | (Group J: 3)<br>Housing, Non-Bank Finance, and Banking. | (Group K: 14)<br>Extractive, Electrical Equipment, Electronics, Steel, Defense, Chemistry, Domestic appliance, Automobile, Light Manufacturing, Food, Telecommunication, Leisure Service, Medical and Nonferrous Metal. |
| **Rm-Rf** | (Group L: 28)<br>All industries | |

**Table 6.** Trading strategies based on condition of factors.

| Condition | When Factor Is Positive | | When Factor Is Negative | |
|---|---|---|---|---|
| Strategies | BUY | SELL | BUY | SELL |
| SMB | Small MV companies in Group A Big MV companies in Group B | Big MV companies in Group A Small MV companies in Group B | Big MV companies in Group A Small MV companies in Group B | Small MV companies in Group A Big MV companies in Group B |
| RMW | Robust ROE companies in Group C Weak ROE companies in Group D | Weak ROE companies in Group C Robust ROE companies in Group D | Weak ROE companies in Group C Robust ROE companies in Group D | Robust ROE companies in Group C Weak ROE companies in Group D |
| HML | High B/M companies in Group E Low B/M companies in Group F | Low B/M companies in Group E High B/M companies in Group F | Low B/M companies in Group E High B/M companies in Group F | High B/M companies in Group E Low B/M companies in Group F |
| CMA | Conservative (low growth rate of assets) companies in Group G Aggressive (High growth rate of assets) companies in Group H | Aggressive companies in Group G Conservative companies in Group H | Aggressive companies in Group G Conservative companies in Group H | Conservative companies in Group G Aggressive companies in Group H |
| CRMHL | High CR companies in Group I | | | High CR companies in Group I |
| AMLH | Low Asset turnover companies in Group J High Asset turnover companies in Group K | High Asset turnover companies in Group J Low Asset turnover companies in Group K | High Asset turnover companies in Group J Low Asset turnover companies in Group K | Low Asset turnover companies in Group J High Asset turnover companies in Group K |
| Rm-Rf | All companies (Group L) | | | All companies (Group L) |

Here we can take an example to illustrate the above table, when SMB is positive, it means in this market, small market value (MV) companies outperform big companies, thus for those industries which is significantly positive correlated with SMB, we should buy small MV stocks and sell large MV stocks. However, for those industries which is negatively correlated with SMB, the strategy should purchase large MV stocks and sell small MV stocks. When SMB is negative, vice versa.

According to the significant level and correlation effect between 7 factors and 28 industries, the trading portfolio and strategy can be conducted. Since the strategic making depends on the positive or negative condition of seven factors, if we can forecast the conditions of seven factors, the trading strategies can be easily made. Thus, for the next two parts, we explored the fluctuation of seven factors to answer two questions:

1. From analysis of factors' changing pattern, can we find the reasons or elements to illustrate the fluctuation of seven factors? (if we find out the driving factor which stimulate the moving of other factors, we can explain the most essential ratio that investor may focus on.)
2. Can we find an approach to forecast the moving of each factor and apply it to do the back test for trading strategies?

*4.2. Time-Series Analysis for Risk Factors*

4.2.1. Endogeneity and Exogeneity for Factors' Cyclical Research

- **Endogeneity**

According to Pearson Correlation matrix, last term data of SMB, RMW, HML, CMA, CRMHL, and AMLH has significant correlation with this term at the level of 0.01. Nevertheless, for the market factor (Rm-Rf), its last term data are nearly independent with this term, which means except market risk factor, other factors are possible to forecast themselves with last term data.

However, to test the endogeneity of these factors, chi-square test are necessary for examination and application of these relationships. After the test, the results show that only SMB, RMW pass the test with 99% confidence level and CMA also has a significance level of 0.05, which means these three factors exist endogeneity and can be truly used to forecast their direction and investment cycle.

- **Exogeneity**

Based on Pearson correlation matrix, the fluctuation of HML, CRHML, and AMLH depend upon the history data of other factors. For details, HML is related to the AMLH of last term. CRHML has a small correlation with HML AMLH bridge its relationships with RMW of last term. Finally, with respect to Rm-Rf, its volatility is partly connected to last term's RMW and AMLH.

Figure 7 summarizes the endogeneity and exogeneity for predicting direction of seven risk factors. In the chi-square test, RMW, SMB, and CMA factor can be only predicted by the last term direction of themselves. With confidence level of 95%, CRHML's direction can forecasted by its history data with HML and the increase and decrease of HML can be predicted by last term direction and AMLH at 90% confidence level. For market factor (Rm-Rf)—even if single stock history data cannot predict next term—with RMW and AMLH, market factor can estimate the direction of its next term with 95% confidence. With a confidence level of 90%, AMLH's fluctuation direction can forecast by RMW. The details of prediction are presented in Appendix D.

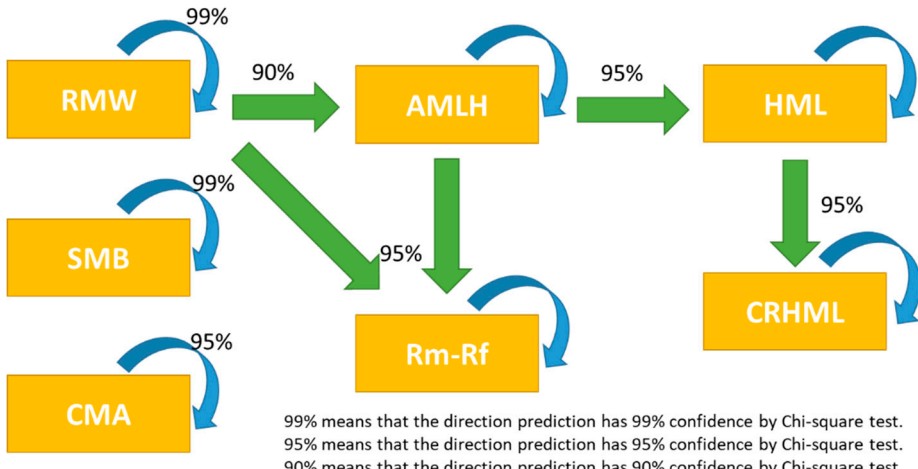

**Figure 7.** Summary of endogeneity and exogeneity for predicting direction of seven risk factors.

### 4.2.2. Trading Strategy with Trend Analysis

In the last two parts, with the examination of chi-square test, the directions of fluctuation of seven factors are predictable, yet the range of fluctuation is still unknown.

Therefore, in order to forecast the positive and negative levels of each risk factor, trend analysis is conducted.

The above seven graphs in Figure 8 describe the trend of each factor. The orange line is the four-season moving average. SMB, HML, and CMA are mostly below 0. By contrast, RMW and CRMHL are above 0. While AMLH and Rm-Rf fluctuate around 0.4.

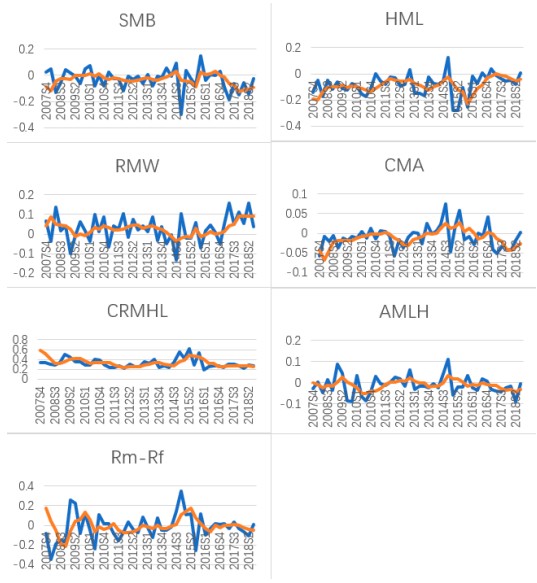

**Figure 8.** Four seasons moving average of seven factors.

### 4.2.3. Back-Testing

Table 7 illustrates the results of the back-testing from 2007S4 to 2018S3 for the trading system. With the strategy, we can achieve positive annual return in SMB, CMA, CRMHL, and AMLH for 15.43%, 3.23%, 29.13%, and 8.4% respectively. To some extent, the result proved the practicability of the seven-factor model, especially for the factor CRMHL in terms of trading strategy. Consequently, with alternative trading strategies which can predict the factor fluctuation more precisely, we can witness a greater margin of improvement.

**Table 7.** Back-testing float return from 2007S4 to 2018S3 and expected annual return for factors.

| Factors | Float Return | Expected Annual Return |
|---------|--------------|------------------------|
| SMB | 165.89% | 15.43% |
| RMW | −6.525% | −0.61% |
| HML | −22.29% | −2.07% |
| CMA | 34.67% | 3.23% |
| CRMHL | 313.09% | 29.13% |
| AMLH | 90.26% | 8.40% |

## 5. Discussion

### 5.1. Analysis of Multi-Factor Model

We compared three multi-factor models in OLS regression and proceed to ridge regression for higher significance and explanatory power. In the seven-factor model, only RMW and HML factor have negative effects on the stock excess return, which is deviated from the assumption. To be specific, negative RMW means that a company with lower ROE can achieve higher excess return than higher ROE company. Also, negative HML means lower book-to-market ratio result in higher excess return.

According to the previous empirical studies in Chinese stock market, the low level of ROE often corresponds to low stock price. On the contrary, relatively high ROE means that the stock price has reached a periodic top. We can infer that stocks with relatively low ROE can attract investment which can boost the stock price. As for the CMA, a company with lower growth of investments has higher excess return. We can infer that investors prefer purchase stock with lower growth of investment.

### 5.2. Industry Analysis

We merely discussed the industries with more than four significant factors. Since they are consistent with the model to the higher extent. As we can see in Figure A10 in Appendix C, among 12 industries, there are four groups of coordinated industries and one group of deviated industry. We found out that the coordinated industries are usually cyclical industries which are highly related to the economic wave. Cyclical industry primarily consists of two categories, resources and industrial raw material. They are closely related to macro economy cycle and supply–demand relationship such as manufacturing, automobile, metal and chemistry industry. Moreover, the coordinated industry is distributed at upstream and downstream; for example, electronics is the upstream industry for telecommunication. Thereby, the chain effect is transmitted from the upstream industry through a series of companies all the way to the downstream industry.

To be specific, regarding different factors, industries may react differently in terms of positive and negative relationship. For SMB, it has a negative effect on steel industry, while six industries stand in the opposite position including electrical equipment, electronics, computers, construction material, telecommunication, etc. There are two reasons for the adverse effect in steel industry. Firstly, steel industry is widely regarded as traditional industry. Secondly, it is dominated by a few giant companies which are supported by the government. Whereas the six companies with positive effects are in opposite condition. They are relatively small in terms of market value. Most companies in electrical equipment and computer industry are in early stage which have potential to achieve higher excess return.

As for RMW, over half of the industry are in a negative relationship, while banking is positive. Since banking is closely related to the macro-economic policy. Therefore, it responds to market situations like ROE more swiftly. Moreover, we ran the regression on the same period which may reveal the lag of the information dissemination in some of the industry. Since the market is imperfect, the companies may not respond to the information in the exact same period.

For HML, over 70% of the companies have a negative effect, while banking has a positive relationship. A higher book-to-market ratio reflects that a bank is increasing the amount of loan issue. Since the major source of incomes of banking come from loan issue, thus a bank with higher

book-to-market ratio can achieve a higher excess return. While other industries with low book-to-market ratio can achieve higher excess return.

As for CMA, five companies—including textiles & garments, utilities, and light manufacturing—consist of manufacturing processes. They have constant cash flow to maintain daily operation and dividend payment. They are also relatively conservative in investment. However, banking and non-bank finance is negative related to CMA factor. It can be inferred that they are more aggressive in investment to achieve higher excess return.

For CRMHL, none of the companies are in negative relationship, while over 70% of the industries are positive to the momentum factor. This means that the momentum effect widely applied to industries, which can result in the similar movement in the next period.

When it comes to AMLH, the housing and finance industries are in a positive relationship. In these industries, assets in terms of land and money reserves are placed in the foremost position, which means that lower asset turnover rates can result in higher excess return. However, for industries with a negative relationship, since asset turnover rate is the reflection of operating capacity, a higher asset turnover rate can lead to higher excess return.

### 5.3. Factor Cyclical Research

According to the result of chi-square test, the future direction of SMB and CMA can be merely predicted based on the previous period. While RMW has the initial effect to the rest of the factors, this chain effect conveys through AMLH and HML to Rm-Rf and CRMHL respectively. Although the size of this effect cannot be predicted, we can find the pattern of investment in this chain effect which can reflect investor behavior in the stock market. Firstly, investors primarily focus on the ROE of the companies. Together with the asset turnover rate, we can predict the future direction of the market premium (Rm-Rf). Secondly, the future direction of CRMHL can also be predicted by HML and AMLH. It can suggest that investors usually refer to the profitability and operation capacity of companies before making investment. These two factors can affect the company evaluation and eventually reflected on the change of momentum factor. According to the investment pattern, we can establish a corresponding investment strategy.

Since the direction of market premium factor (Rm-Rf) can be predicted, to some extent, we can find out the pattern of the index. Therefore, it can be also be applied in the investment of index futures and options through call and put.

### 5.4. Trading Strategy and Back Test

According to the trend analysis, a moving average model is applied in the model (Hanke and Wichern 2014). However, the span of moving average is not specifically decided, which can affect the result of back-testing. We only conducted a four-season moving average as an example. In order to find out the optimal moving average, iteration can be applied to figure out the optimal trading strategies.

### 5.5. Significance and Limitations of Research

With the comparison of previous work and our research, academic significance can be illustrated in two aspects.

For one thing, prior studies mainly focus on the effect of factors on excess return from a portfolio aspect. Stocks are classified into different groups based on their characteristics. For example, the companies are divided into two groups based on size, then each group is divided into two groups based on the book-to-market ratio. By comparing the average return of the four groups, they can find out the relationship between factors and return. Thus, their methods can only be applied to investments of a specific portfolio, which may lack of practicability and explanatory power in application to a single stock or specific industries. For this research, nevertheless, we focus on discovering different effects of risk premium factors to the excess return of single stock via OLS and ridge regression and then summarize the significant correlations between factors and different industries, which means our

research is more practical. For example, based on our model, investors can judge the factors which significantly affect a specific stock or industry.

Additionally, previous works have never discussed the risk premium factors from the aspect of time-series analysis. While forecasting the investment cycle, prior research only investigated from a macro-economic level or technical analysis. For one example, in *Business Cycles* (Lars 2006), the author merely analyzed the cycle of housing, credit, and inventory, which belonged to macro-economic level. For another example, technical analyses, like Elliott wave principle and candlestick charts, aim to describe the market price wave pattern on time series. Thereafter, there is an academic gap of discussing investment cycle of fundamental analysis. In this paper, however, the fluctuations of seven factors are studied by chi-square test of endogeneity and exogeneity of factors. Moreover, it is a new type of cycle analysis because it explains the companies' value behind the investment decision.

The contribution consists of four points. Firstly, we applied a novel method to find out the correlations between seven risk factors and each single stock. Secondly, we find out and explain the correlations between seven risk factors and each industry and specify the situations (positive or negative) of risk factors to buy or sell the industries' stocks. Thirdly, the result of cyclical analysis can be applied in forecasting the direction of risk factors, especially the market risk factor (Rm-Rf) which can be used in transaction of options and futures of market index. Lastly, a back-test was conducted in a simple trading system which suggested that SMB (size premium), CMA (investment growth premium), CRMHL (momentum premium), and AMLH (asset turnover premium) can gain positive returns.

As for the limitations, this research conducted a four-seasons-moving average method to forecast the level of seven factors. Even if it proved that SMB, CMA, CRMHL, and AMLH can be applied in trading system. However, more forecasting methods like ARIMA, VAR, and ANN can be constructed to form better trading strategies. More limitations are presented with our direction for further study.

## 6. Conclusions and Further Study

In the research of relationship between risk premium and excess return, the hybrid approach takes a primary position. In the examination and modification of multi-factor model, OLS and ridge regression are conducted on Models 1 to 4. The seven-factor model has an optimal combination of p level and R square. In chi-square test, the effect of factor was responded differently in each industry. Therefore, p level was calculated based on the industry in order to find out the well-fitted coordinated and deviated industry. Bank and steel industry are well-fitted in the model, while industries within the same stream—e.g., telecommunication and electronics—are found to be coordinated industries. Moreover, cyclical industry is usually coordinated except for banking.

In the factor cyclical research, chi-square and correlation tests are applied to find out the endogeneity and exogeneity. SMB, RMW, and CMA have endogeneity while the remaining four factors have both endogeneity and exogeneity. With the pattern of direction in factors, an investment strategy was established. To be specific, when moving average SMB is positive, small market value (MV) companies outperform big companies, thus for those industries which are significantly positively correlated with SMB, it would be better to buy small MV stocks and sell large MV stocks. With the strategy, we can achieve a positive return in SMB, CMA, CRMHL, and AMLH respectively in the back test.

As for the further study, the proportion of individual investors in the Chinese stock market contribute nearly 80% of the trading volume. Therefore, investor preference and the irrational behavior should be considered. The investor sentiment factor may be added to improve the explanatory power.

Previous studies also illustrated that the reversal effect exists in the long-term stock market. If our period is set to be longer than one year, we can add the reversal factor to explain the reversal effect.

Dynamic analysis can be applied to data processing, since the economic environment changes over time. We can also monitor the stock market and provide suggestion in stock selection which can fit for the target return.

Moreover, we can further study the inter-factor drive relationship, in order to establish more investment strategies. To be specific, by optimizing the back-test ratio and Sharpe ratio, we can construct a better investment portfolio.

**Funding:** This research received no external funding form government or any institution.

**Conflicts of Interest:** We declare that there is no conflict of interest in this research.

## Appendix A. Intuition and Assumption Behind the Hypotheses

There is a trade-off between return and risk. In order to find the corresponding risk for the excess return, according to CAPM model, there is a market risk. To illustrate the abnormal return, Fama used three-factor model which added size premium and book-to-market premium in 1993. In 2013, in their five-factor model, profitability and investment growth are also considered to be significant coefficients. Next, the intuition behind these premiums will be explain one by one.

1.   Market Premium

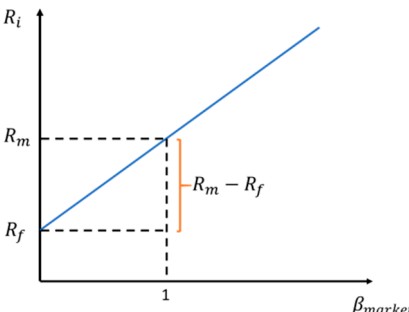

**Figure A1.** Positive relationship between expected return and market premium.

Market premium is represented by the difference between market return and risk-free rate. Since the fluctuation of stock market's expected return is higher than the risk-free rate, namely stock market has higher risk, the expected return of stock market should higher than risk-free rate.

2.   Size Premium

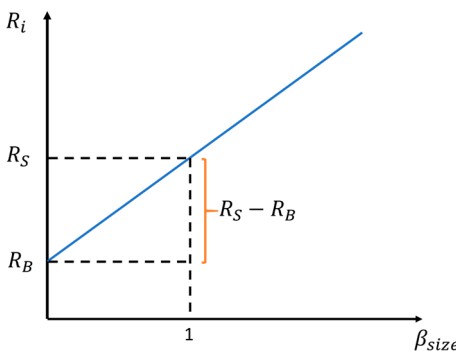

**Figure A2.** Positive relationship between expected return and size premium.

Size premium is the difference between small companies' average return and big companies' average return in a diversified portfolio or in the market. Normally, small companies have higher returns than the bigger ones. Because at the same period of time, small companies' profit is easy to grow faster than big companies and the growth rate of their dividend also is higher.

According to the DDM model, the price of security depends on the discounted present value of future dividend. In this formula, 'P' is the expected present price of one stock. '$D$' is the dividend of

this year and '$g$' is the constant growth rate of the dividend (it also is the growth rate of profit). For the last, '$i$' is the dividend interest rate, which may refer to risk-free rate.

$$\begin{aligned}
P &= \lim_{n\to\infty}\left[\frac{D(1+g)}{1+i} + \frac{D(1+g)^2}{(1+i)^2} + \frac{D(1+g)^3}{(1+i)^3} + \dots + \frac{D(1+g)^n}{(1+i)^n}\right]\\
&= \lim_{n\to\infty}\left[\frac{D(1+g)}{1+i} * \frac{\left(\frac{1+g}{1+i}\right)^n - 1}{\frac{1+g}{1+i} - 1}\right]
\end{aligned}$$

Given by $i > g$,

$$\left(\frac{1+g}{1+i}\right)^n \to 0$$

So, $P = \frac{D(1+g)}{i-g}$

$$\therefore P_1 = \frac{D_1(1+g_1)}{i-g_1}, \quad P_2 = \frac{D_2(1+g_1)}{i-g_1}$$

If we assume $D_1 = D_2$

Hence, the return rate of the stock can be represented by

$$R_i = \frac{P_2 - P_1}{P_1} = \frac{(g_2 - g_1)(i+1)}{(1+g_1)(i-g_2)}$$

In terms of small companies and big companies, $g_1$ (*the growth rate of first term*) *and i* (*risk − free rate*) *are the same.*

However, in the second term, with the control of other effects, growth rate ($g_2$) of small companies is higher than the big one, so the expected return ($R_i$) of small companies is generally higher than the large one.

In other words, high returns of small companies indicate they also carry higher risk. If investors can suffer the risk brought by small companies, they can gain the risk premium which is called 'size premium'.

3.   Book-to-Market Premium

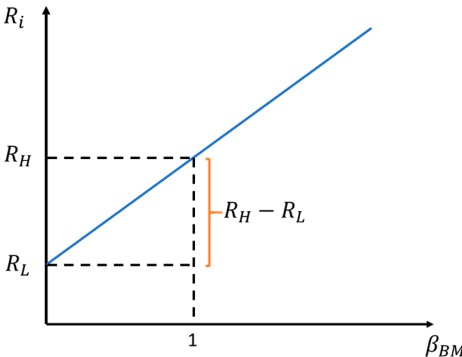

**Figure A3.** Positive relationship between expected return and book-to-market premium.

Book-to-market premium is the difference between high B/M ratio companies' average return and low B/M ratio companies' average return in a diversified portfolio or in the market.

There is an effect named B/M effect which indicates that higher B/M ratio companies has higher excess return. It can be illustrated by prospect theory easily. However, in this case, the $x$-axis is MV and the $y$-axis is the value.

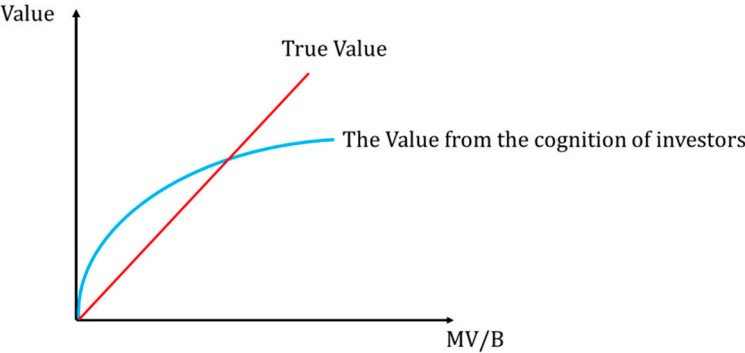

**Figure A4.** Relationship between true and cognitive value of stocks with the growth of MV/B.

For the higher B/M ratio companies, the MV/B is relative lower and people always overprice the true value of a stock, it leads to the demand for those companies is higher. With the growth of demand and stock price, the return of those stocks also is higher.

Vice versa, higher B/M ratio companies has lower expected return and excess return (expected return minus risk of free rate).

Therefore, if the investor prefers higher B/M ratio companies, they take the risk of B/M effect on one hand, they gain the B/M premium on the other hand.

4. Profitability Premium

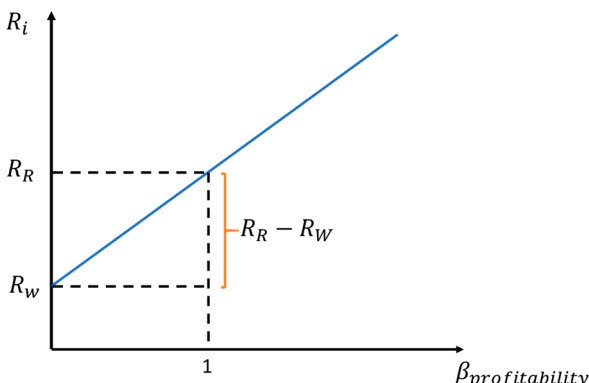

**Figure A5.** Positive relationship between expected return and profitability premium.

Profitability premium is the difference between robust profitability (higher ROE) companies' average return and weak profitability (lower ROE) companies' average return in a diversified portfolio or in the market.

With the control of other effects, companies with robust profitability—measured by the level of ROE (return of equity)—outperform in their expected rate of return and take greater variance. This is because companies with high profits also distribute high dividends.

$$P = \frac{D(1+g)}{i-g}$$

According DDM (dividend discount model) formula, higher dividend means higher price and demand which will enhance the level of expected return. If an investor purchases companies with robust profit, they may get higher excess return and fluctuation at the same time. Vice versa, weak profitability companies bring people low excess return and risk.

5.   Investment Growth Premium

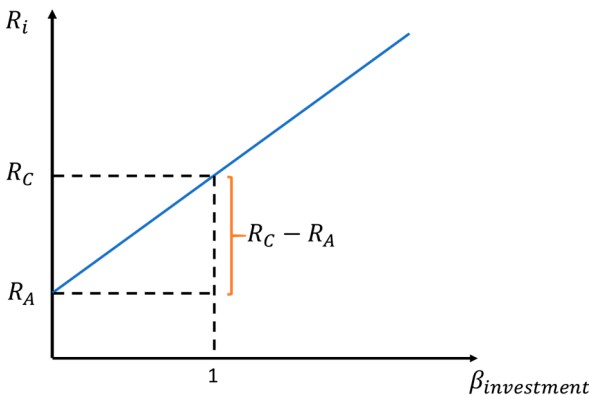

**Figure A6.** Positive relationship between expected return and investment growth premium.

Investment growth premium is the difference between conservative (lower growth rate of investment or lower growth rate of assets) companies' average return and aggressive (higher growth rate of investment or higher growth rate of assets) companies average return in a diversified portfolio or in the market.

The reason why aggressive companies may have low excess return and risk is that these kinds of firms allocate more profit into reinvestment rather than dividends, thus it decreases the expected price and return, which leads to low risk. Vice versa, conservative companies bring higher excess return and risk because of larger amount of dividend rather investment.

6.   Momentum Premium

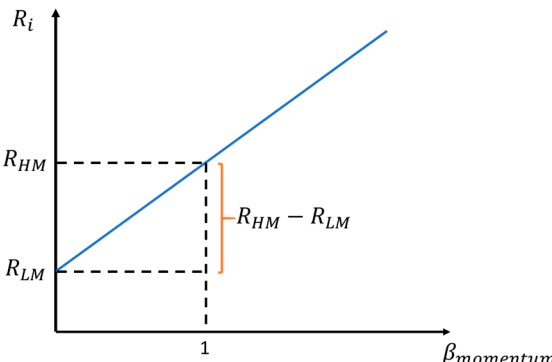

**Figure A7.** Positive relationship between expected return and momentum premium.

Momentum premium is the difference between higher momentum (higher accumulated return) companies' average return and lower momentum (lower accumulated return) companies' average return in a diversified portfolio or in the market.

Momentum is the accumulated return in one quarter. The higher one means the stock is popular with high return and risk. Vice versa, people invest in low momentum companies with low premium and risk.

7.　Asset Turnover Premium

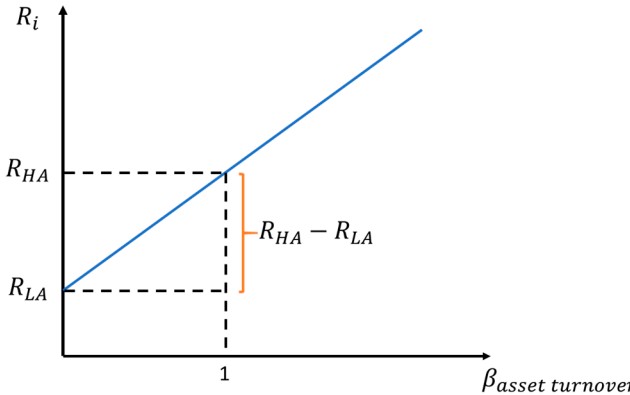

**Figure A8.** Positive relationship between expected return and asset turnover premium.

Asset turnover premium is the difference between higher asset turnover companies' average return and lower asset turnover companies' average return in a diversified portfolio in the market.

Asset turnover is the total revenue divided by total asset. The higher one means the stock is popular with high return and risk. Vice versa, people invest in low momentum companies with low premium and risk. The regression analysis was conducted with a opposite direction.

**Appendix B. Chi-Square Test of Industry in Different Factors**

Table A1 recorded the total chi-square value of 28 industries for seven factors. By conducting the chi-square test, we can find out the different effect of factors to various industries. When the total chi-square value is larger than the critical value (when df = 27), it means that each industry is responding differently to the specific factors. According to the result, six factors passed the test, while Rm-Rf is insignificant. Therefore, we need to discuss the effect of factors based on different industry.

**Table A1.** Chi-square test of industry in different factors.

| Factors | Chi-Square ($\chi^2$) |
|---------|----------------------|
| SMB | 97.58155 |
| RMW | 102.9963 |
| HML | 134.683 |
| CMA | 85.7257 |
| Rm-Rf | 38.17769 |
| CRMHL | 66.58794 |
| AMLH | 179.6194 |

**Appendix C. Significance Level and Correlation Effect**

**Table A2.** Significance level of factors in different industries.

| Title | SMB | RMW | HML | CMA | CRMHL | AMLH | Rm-Rf |
|-------|-----|-----|-----|-----|-------|------|-------|
| Extractive | 0 | −0.76 | −0.8 | −0.72 | 0.72 | −0.72 | 1 |
| Media | 0 | −0.78571 | −1 | 0 | 0 | 0 | 1 |
| Electrical equipment | 0.741935 | −0.80645 | −0.93548 | 0 | 0 | 0 | 1 |
| Electronics | 0.833333 | 0 | −0.875 | 0 | 0.729167 | −0.70833 | 1 |
| Housing | 0 | 0 | −0.72527 | 0 | 0.714286 | 0.791209 | 1 |
| Textiles & garments | 0 | −0.70833 | −0.875 | 0.708333 | 0.916667 | 0 | 1 |
| Non-bank finance | 0 | 0 | 0 | 0 | 0 | 0.714286 | 0.964286 |
| Steel | −0.73684 | −0.94737 | 0.789474 | 0 | 1 | −0.89474 | 1 |

**Table A2.** *Cont.*

| Title | SMB | RMW | HML | CMA | CRMHL | AMLH | Rm-Rf |
|---|---|---|---|---|---|---|---|
| Utilities | 0 | 0 | 0 | 0.710526 | 0.828947 | 0 | 1 |
| Defense | 0 | −0.8 | −0.72 | 0.8 | 0.8 | −0.84 | 1 |
| Chemistry | 0 | −0.74699 | −0.78313 | 0 | 0.795181 | −0.84337 | 1 |
| Mechanical equipment | 0 | −0.76667 | −0.76667 | 0 | 0.833333 | 0 | 1 |
| Computer | 0.777778 | 0 | −0.92593 | 0 | 0.777778 | 0 | 1 |
| Domestic appliance | 0 | 0 | 0 | 0 | 0.857143 | −0.7619 | 1 |
| Construction material | 0.714286 | −0.7619 | 0 | 0 | 0.904762 | 0 | 1 |
| Construction ornament | 0 | 0 | 0 | 0 | 0.933333 | 0 | 1 |
| Transportation | 0 | −0.81633 | 0 | 0 | 0.857143 | 0 | 1 |
| Animal husbandry and fishery | 0 | 0 | −0.71429 | 0 | 0 | 0 | 1 |
| Automobile | 0 | −0.86364 | −0.75 | 0 | 0.818182 | −0.75 | 1 |
| Light manufacturing | 0 | −0.73077 | −0.76923 | 0.730769 | 0.730769 | −0.80769 | 1 |
| Commercial | 0 | 0 | −0.88525 | 0.754098 | 0 | 0 | 1 |
| Food | 0.794118 | 0 | 0 | 0 | 0.794118 | −0.76471 | 1 |
| Telecommunication | 0.727273 | 0 | −0.77273 | 0 | 0.772727 | −0.77273 | 1 |
| Leisure service | 0.8125 | −0.875 | −0.8125 | 0 | 0.8125 | −0.8125 | 1 |
| Medical | 0 | 0 | −0.80412 | 0 | 0.835052 | −0.74227 | 0.989691 |
| Bank | 0 | 0.714286 | 1 | −0.71429 | 1 | 1 | 1 |
| Nonferrous metal | 0 | −0.88372 | −0.74419 | −0.83721 | 0.790698 | −0.86047 | 1 |
| Comprehensive | 0 | −0.92308 | −0.80769 | 0 | 0 | 0 | 1 |

In this table, 0 represents that the factor is insignificant to the industry. The absolute value represents the percentage of the significant coefficients. A positive number represents that the factor has positive effect on the industry, while negative number represents that the factor has a negative effect on the industry.

## Appendix D. Forecasting the Direction of Factors

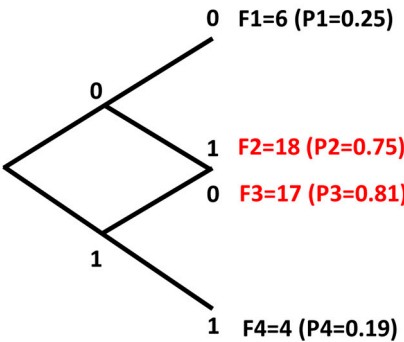

**Figure A9.** Use SMB to forecast the direction of SMB.

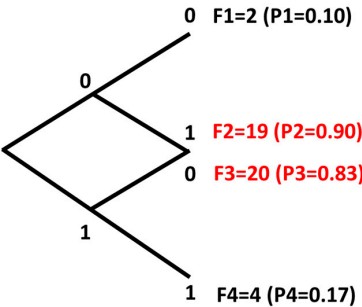

**Figure A10.** Use RMW to forecast the direction of RMW.

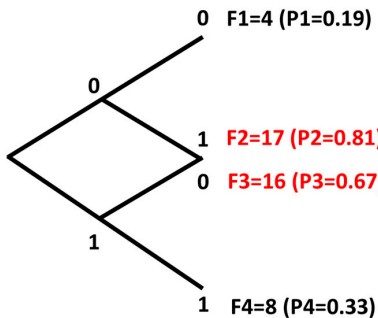

**Figure A11.** Use CMA to forecast the direction of CMA.

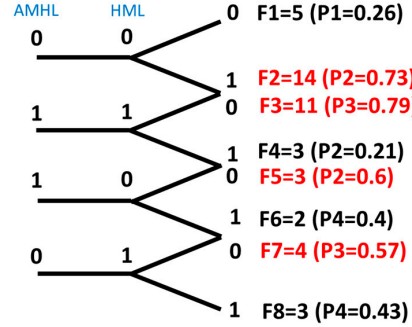

**Figure A12.** Use AMLH and HML to forecast the direction of HML.

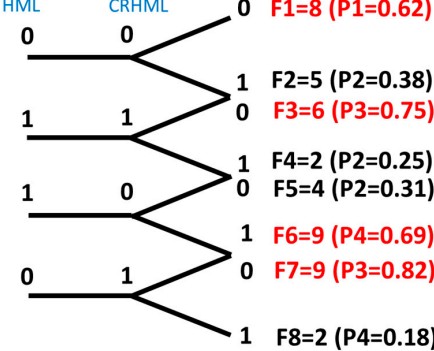

**Figure A13.** Use HML and CRMHL to forecast the direction of CRMHL.

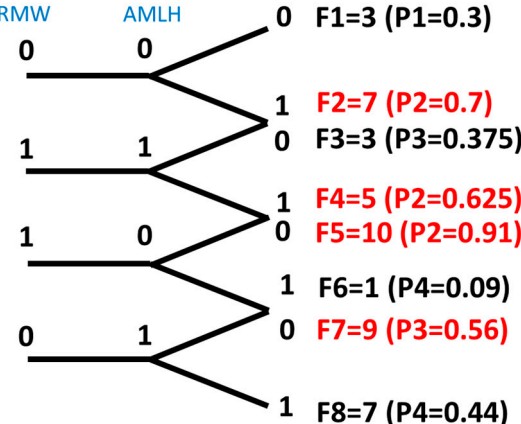

**Figure A14.** Use RMW and AMLH to forecast the direction of AMLH.

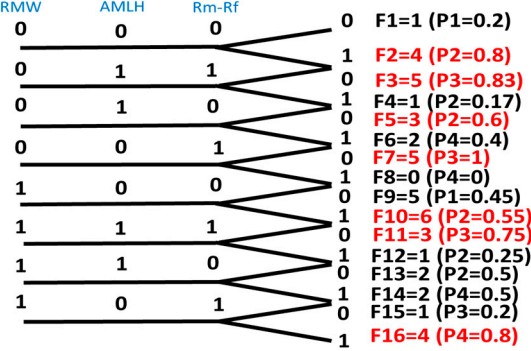

**Figure A15.** Use RMW, AMLH, and Rm-Rf to forecast the direction of Rm-Rf.

## Appendix E. Result of Stability Test

```
p_level= 0.7848678213309025
p_level= 1.0
p_level= 1.0
p_level= 1.0
p_level= 1.0
p_level= 1.0
p_level= 1.0
p_level= 0.6244302643573382
p_level= 0.11030082041932543
p_level= 0.2506836827711942
```

**Figure A16.** Significance level of variables.

The figures in the red frame are the significance level of time, time^2 and season.

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
