# Peer review of "Examination and Modification of Multi-Factor Model in Explaining Stock Excess Return with Hybrid Approach in Empirical Study of Chinese Stock Market"

_jrfm, doi:10.3390/jrfm12020091_

Round 1

Reviewer 1 Report

In terms of a general view, the manuscript covers an interesting topic and manages to provide a relevant research question oriented in the direction of multi-factor models construction. However, the manuscript has a series of flaws that should be addressed:

1.     The introduction section is weak and should be rewritten. I advise the authors to provide separate sections for the introduction and the literature review. The 1.1 section should be extensively enriched in order to show the gap in literature that the authors are considering and the main contributions brought by the manuscript. It should also incorporate section 1.3 into a coherent discussion. A strong focus should be given to highlighting the original contributions in relation to past literature.

2.     The literature review should come self-standing, written in a clear way that is not sequential. I do not consider that breaking it into subsections will add value to the manuscript. It will only make the manuscript more truncated and highlight its problems in terms of writing, cursivity and presentation. I am also not sure if equations (1) and (2) are relevant in such a section and in the general context of the manuscript.

3.     Section 2.1 should be enriched with comments. Enumerating a list of hypotheses rarely does enough in terms of clarity.

4.     Figure 1 is redundant. It should be removed and replaced by a clear text (written in good English) that explains the methodological construction. This could be obtained by rewriting section 2.2 and 2.3 in a coherent way.

5.     Figure numbering jumps from Figure 1 (page4) to Figure 3 (page 7) directly.

6.     Figure 3 has limited relevance and could be removed.

7.     Section 2.3 is written in a very bad form of English and should be heavily upgraded.

8.     Figure 2 appears on page 9. Again, it could be removed due to limited relevance. Comments are more suitable.

9.     Section 2.6 is simplistic and redundant. I do not think that it adds value to the manuscript given the fact that such aspects are common knowledge even for the most uninformed readers and followers of the journal.

10.  The results section is poorly constructed. It is littered with output and contains little to no comments.

11.  Table 8 (page 17) is a Figure and not a table. It las limited relevance. It should be replaced by well-written text.

12.  Breaking section 4 Discussion into numerous small components might not be the best idea. The entire section requires heavy language updates.

13.  I encourage the authors to try and perform a better handling of appendices, as this is terribly done at present. The information should be filtered for relevance, as maybe certain aspects can be removed. It might a good idea to bring several elements to the body of the article and drop others.

All in all, despite an interesting idea the manuscript is badly handled, composed and written, making it fall short of publication standards.

Reviewer 2 Report

The topic of the paper is interesting and it could be a valuable contribution to the journal, given the focus of many academics on the Chinese stock exchanges. The design of the research is mostly appropriate and the methodology of the study is sound. Nonetheless, the paper requires some substantial changes in terms of the presentation of the results as well as in-depth proofreading. Below I provide my detailed comments.

Abstract should be revised by adding information about the time period of the analysis and summary of the results.

I suggest expanding the descriptions of the models in the Introduction through addition of their equations - in each case the expected (required) rate of return should be on the left side (equations such as no. 1 provided in the paper are difficult to grasp, in particular for the reader unfamiliar with these concepts). Other additions should include more details about arbitrage pricing theory, more empirical studies on the presented models and identification of the empirical studies mentioned in the last paragraph of Sect. 1.2.4.

Authors write that individual investor account for 'barely' 80% of the total turnover - in comparison to the other markets that share is rather high.

Hyptheses should be followed by detailed explanation of some of the utilized concepts as they may be ambiguous (e.g., profitability premium).

More details about the database should be provided in the paper.

Authors write that they included in their dataset data on the companies that conducted their IPOs in the recent years but later they mention that time series with missing observations were deleted. These statements seem contradictory.

Panel models cannot be estimated for each stock as this violates the basic idea of panel modelling - I understand the concept of the authors but it should be reformulated.

Many parts of the methodology and presentation of the results are highly unclear. Examples include Sect. 2.3.1. or following 2.3.2.

Sentence in the line 203 is poorly written and unclear.

'Robust test' - in my opinion it should be 'Robustness test'.

Title of the Sect. 2.4. must be expanded.

Market premium in Table 1 is poorly explained - why do authors suddenly refer to the exchange market index?

The sentence in the lines 345-346 is difficult to understand - histrograms can be used to assess the distribution even when it is not normal as in this case.

Line 351: 'uncorrected' - I guess it should be 'uncorrelated'.

Line 398: I do not understand the meaning of 'deviated'

Analysis of the banking sector in Sect. 4.2. should be revised and improved because it includes some vague and partially incorrect statements.

Contribution of the paper in Sect. 4.5. needs to be rewritten, in particular the second paragraph is almost entirely unclear.

Line 600 (and other parts of the paper): it should be 'Sharpe ratio'.

List of references should be alphabetical (see the last position). Moreover, this is an intensively studied field and more publications should be used.

Derivation of the GMM model and some graphs in App. 1 are not necessary.

Round 2

Reviewer 1 Report

The present version of the manuscript is slightly better than the original, despite being not very close to my suggestions. Despite this fact, I acknowledge the efforts of the authors in trying to obtain a more refined paper. Please find bellow my comments on the author input. 

Comment 1 and Comment 2: The upgrades and restructuring in the introduction and literature review are a good add-on.

Comment 3 Hypotheses description is a welcomed plus.

Comment 4. I can agree with the comment made by the authors. However, I could not locate the text to which they refer.

Comment 7: Could not find the upgrades suggested by the authors.

Comment 5, Comment 6, Comment 8 – Formatting has been conducted.

Comment 9 and Comment 10. I acknowledge the changes and agree to the comments provided by the authors.

Comment 11: Formatting has been conducted.

Comment 12 and 13 – I am not sure I agree to this logic, but it is a matter of personal taste.

All in all, the manuscript is in better shape than the first version and could be considered for publication, despite several minor impediments.

Author Response

The present version of the manuscript is slightly better than the original, despite being not very close to my suggestions. Despite this fact, I acknowledge the efforts of the authors in trying to obtain a more refined paper. Please find bellow my comments on the author input. 

 Point 1:Comment 1 and Comment 2: The upgrades and restructuring in the introduction and literature review are a good add-on.

Response 1: Thank you so much for your agreement and the suggestions you provided for us before.  

Point 2:Comment 3 Hypotheses description is a welcomed plus.

Response 2: Thank you so much for your agreement and the suggestions you provided for us before.  

Point 3:Comment 4. I can agree with the comment made by the authors. However, I could not locate the text to which they refer.

Response 1: Thank you so much for your agreement and the suggestions you provided for us before.  The description for Figure 1 is rewrote in a clear way.

Point 4:Comment 7: Could not find the upgrades suggested by the authors.

Response 1: Thank you so much for your agreement and the suggestions you provided for us before.  The original section 2.3 is section 3.3 now. We have modified a lot of grammar mistakes.

Point 5:Comment 5, Comment 6, Comment 8 – Formatting has been conducted.

Response 1: Thank you so much for your agreement and the suggestions you provided for us before.  

Point 6:Comment 9 and Comment 10. I acknowledge the changes and agree to the comments provided by the authors.

Response 1: Thank you so much for your agreement and the suggestions you provided for us before.  

Point 7:Comment 11: Formatting has been conducted.

Response 1: Thank you so much for your agreement and the suggestions you provided for us before.  

Point 8:Comment 12 and 13 – I am not sure I agree to this logic, but it is a matter of personal taste.

Response 1: Thank you so much for your agreement and the suggestions you provided for us before. The logic in the section of discussion is the same as the logic in part of results.

All in all, the manuscript is in better shape than the first version and could be considered for publication, despite several minor impediments.